# Ankle Joint MRI—Comparison of Image Quality and Effect of Sports-Related Stress

**DOI:** 10.3390/diagnostics13172750

**Published:** 2023-08-24

**Authors:** Robert A. J. Gorzolla, Udo Rolle, Thomas J. Vogl

**Affiliations:** 1Department of Diagnostic and Interventional Radiology, University Hospital Frankfurt, Theodor-Stern-Kai 7, 60590 Frankfurt, Germany; rgorzolla@icloud.com (R.A.J.G.); thomas.vogl@kgu.de (T.J.V.); 2Department of Paediatric Surgery and Paediatric Urology, University Hospital Frankfurt, Theodor-Stern-Kai 7, 60590 Frankfurt, Germany

**Keywords:** magnetic resonance imaging, ankle joint, cartilage volume

## Abstract

Objectives: The main aims of the study were the evaluation of stress-related effects (strenuous vs. non-strenuous sport vs. nonathletes) in stimulating or reducing influences on cartilage volume in the ankle joint and the evaluation of the image quality of a magnetic resonance imaging (MRI) device with a field strength of 3.0 Tesla compared to one of 1.5 Tesla. Methods: A total of 15 subjects (6 male, 9 female) aged 19–33 years participated voluntarily in this prospective study. The subjects were divided into three groups: high-performance athletes of the German Football Association (DFB) (football/soccer = strenuous sport), high-performance athletes of the German Swimming Association (DSV) (swimming = non-strenuous sport), and nonathletes. MRI was performed on both ankle joints of all subjects in the 1.5 T and 3.0 T MRI scanners using survey sequences, proton density sequences in the coronal and sagittal planes, and VIBE sequences. Using the images of both feet produced by VIBE sequences, the cartilages of the talus and tibia were manually circumscribed using a computer mouse in every third layer, and the volume was calculated. For qualitative assessment, blinded images were submitted to three radiologists with defined standards. The images were scored using a scale from 1 to 5. Results: Cartilage volume: The investigation and examination of the individual cartilage volumes by analysis of variance (ANOVA) showed no significant differences among the three groups. The effect intensities, as calculated by Cohen’s d, were right tibia (Ti_ri_) = 2.5, left tibia (Ti_le_) = 2.2, right talus (Ta_ri_) = 1.9, and left talus (Ta_le_) = 1.6 in the strenuous sport versus nonstrenuous sport groups; Ti_ri_ = 0.8, Ti_le_ = 1.2, Ta_ri_ = 0.4, and Ta_le_ = 0.5 in the strenuous sport versus nonathlete groups; and Ti_ri_ = 0.3, Ti_le_ = 0.2, Ta_ri_ = 0.7, and Ta_le_ = 0.5 in the nonstrenuous sport versus nonathlete groups. Device comparison: In the investigation of each evaluated area on the 1.5 T and 3.0 T MR images by the Wilcoxon matched-pair test, significant differences were found for the cartilage–bone border (KKG = 0.002), cancellous bone (Sp = 0.001), medial ligamentous apparatus (mBa = 0.001), lateral ligamentous apparatus (lBa = 0.001), and adipose tissue (Fg = 0.002). Thus, there were significant differences in the assessment of the 1.5 T MRI and the 3.0 T MRI in all five evaluated areas. Conclusion: The study showed no significant difference in the volume of hyaline articular cartilage in the upper ankle joint among the high-performance strenuous DFB athlete, high-performance non-strenuous DSV athlete, and nonathlete groups. The 3.0 Tesla device offers significant advantages in image quality compared to the 1.5 Tesla device.

## 1. Introduction

The hyaline articular cartilage of the upper ankle joint is exposed to extensive strain in everyday life [1]. Physical exercise and competitions increase the forces acting on the joint. Physical training increases bone and muscle mass [2], but little is known about changes in the hyaline articular cartilage of the upper ankle joint under stress in vivo. Different studies show different results regarding cartilage volumes under load. However, there is a clear difference in cartilage volume between male and female subjects [3].

In addition to ordinary strain, non-physiological impacts such as distortions also play an important role in cartilage stress [4]. The consequence of excessive strain, especially in the context of injuries, is post-traumatic arthrosis. This could be related to small talocrural cartilage contact areas under load [5]. The primary arthrosis of the ankle joint, however, is rare [6,7].

Imaging plays a major role in the diagnosis of the cartilage changes related to stress and arthrosis. Osteochondral lesions of the ankle could be diagnosed using radiographs, ultrasonography, computed tomography, single-photon emission computed tomography, and magnetic resonance imaging [8]. Magnetic resonance imaging (MRI) is non-invasive and enables the direct visualization of the cartilage as well as cartilage disorders. Furthermore, MRI could also be utilized in the prevention of sports injuries in physical education teaching [9]. The application of MRI-based quantitative parameters of cartilage, i.e., thickness and volume, have been shown to be valuable for the diagnosis and follow-up of athrotic changes in joints in general and the ankle joint in particular. The 1.5 Tesla magnetic resonance imaging (MRI) technique has been applied and shown to be valuable in research and clinical applications to investigate the morphometric features of several different joints [10,11]. The 1.5-Tesla MRI of joint cartilages also allows reproducible measurements for daily usage in diagnosis [12]. Compared to the 1.5 Tesla magnetic resonance imaging (MRI) technique, the 3.0 Tesla MRI technique has good potential to provide higher diagnostic precision in the field of musculoskeletal imaging [13,14,15]. The 3.0 Tesla MRI technique has additionally been shown to provide reproducible measurements of cartilage morphology in a multicenter clinical trial [16].

This study was designed to compare the 3.0 T MRI with the 1.5 T MRI, especially regarding the imaging quality, whereby images blinded by different radiologists will be examined. Here, the delimitability of the bone–cartilage boundary, the representation of the ligament and bone structure, as well as the overall image quality will be examined.

Furthermore, information on the sport-specific influences of athletic training/competition on the cartilage volume of respective joints is crucial [17,18]. Therefore, in the study presented here, a comparison of a strenuous sport group (football) with a non-strenuous sport group (swimming) was undertaken. Sport-specific stresses in football result in additional forces acting on the joints. In particular, the various acceleration and deceleration movements, including changes in direction, play a major role [19]. Furthermore, there are non-physiological forces acting on the joints, e.g., in the context of ankle joint distortions. In particular, the influence of strenuous sports needs to be revealed. The extent to which increased stress exerts a growth stimulus on the ankle cartilage, or whether degenerative changes occur earlier, will also be investigated.

The aim of this study was to compare competitive athletes (strenuous and non-strenuous) with the reference group of non-athletes. Furthermore, information on the general influences of athletic training/competition on the cartilage volume of the upper ankle joint will be obtained by comparing the strenuous sport and nonstrenuous sport groups with the non-athlete group.

## 2. Materials and Methods

### 2.1. Study Design and Patients

A total of 15 subjects (9 female, 6 male) aged 19–33 years participated in this pilot study. This study was undertaken to investigate the suitability of the MRI investigation approach for the evaluation of cartilage volume to establish this investigation as a standard.

The investigation was approved by the clinical ethics committee of the Faculty of Medicine, Goethe University, Frankfurt (No. 26/12).

Individual athletes and non-athletes were included after having signed their consent according to their expected exposition to strain (Table 1).

The subjects were divided into the following three groups:High-performance athletes of the German Football Association (DFB) (5 female). Only athletes with proven squad membership were admitted to the study.High-performance athletes of the German Swimming Association (DSV) (2 female, 3 male). Only athletes with proven squad membership were admitted to the study.Nonathletes (2 female, 3 male). Only subjects who did not engage in any sporting activity or, at most, one sporting activity per week, but who had never engaged in competitive sport, were admitted to the study.

All subjects underwent MRI of both ankle joints in two different devices. In the area of the ankle joints to be examined, musculoskeletal diseases, trauma or pain were excluded at the time of the examination. The inclusion and exclusion criteria were reviewed and contraindications were excluded.

### 2.2. MRI Diagnostics

The examinations alternated between the two MRIs (1.5 T or 3.0 T) using an orthopedic coil. The ankle was fixed in a lower leg vacuum splint in a neutral position. Survey sequences, proton density sequences in the coronal and sagittal planes, and VIBE sequences in the coronal plane were obtained (see Table A1 and Table A2).

Following the first examination, the subjects switched to the other MRI scanner. The quantitative assessment of the MRI data was performed using Volume Renderer. On the images of both feet obtained from the VIBE sequence, the cartilages of the talus and tibia were manually circumscribed using a computer mouse in every third layer, including only load-bearing cartilage domains of the tibia. The system calculated the volume of the bypassed areas. An image of the three-dimensional representation of the talar and tibial articular cartilage was saved with the volume indicated. The same procedure was performed with the data sets of the other foot.

For qualitative assessment, blinded images were presented to three radiologists with specialist standards. The assessment was based on the following scale: sharp (5 points), clear (4 points), relatively blurred (3 points), blurred (2 points), and not recognizable (1 point).

### 2.3. Statistical Analysis

The data analysis was performed using SPSS version 18.0.

The collected data were tested for normality using the Shapiro–Wilk test (suitable for small numbers), among others, and the mean values and the corresponding standard deviations were determined. The investigation of cartilage volume was carried out using analysis of variance (parametric). The Wilcoxon test (non-parametric) was used to assess the significance. Finally, the effect intensities were calculated using Cohen’s d.

## 3. Results

A total of 15 subjects (9 female, 6 male) aged 19–33 years participated.

The average age of the strenuous sport group (*n* = 5) was 29.60 ± 2.19 years (range = 27–33 years). The average weight was 61.60 ± 3.05 kg (range = 58–66 kg). The average height was 169.20 ± 6.45 cm (range = 164–180 cm). The sex ratio w/m was 5/0.

The average age of the nonstrenuous sport group (*n* = 5) was 23.0 ± 4.35 years (range = 19–30 years). The average weight was 76.20 ± 7.79 kg (range = 65–87 kg). The average height was 188.40 ± 8.26 cm (range = 176–198 cm). The sex ratio w/m was 2/3.

The average age of the nonathlete group (*n* = 5) was 25.60 ± 3.36 years (range = 22–29 years). The average weight was 78.40 ± 17.89 kg (range = 62–106 kg). The average height was 178.20 ± 10.89 cm (range = 165–189 cm). The sex ratio w/m was 2/3.

### 3.1. Cartilage Volume

In total, the cartilage volumes of the right and left ankle joints of 15 subjects were examined. The average age of all subjects (*n* = 15) was 26.07 ± 4.23 years. The youngest subject was 19 years old, and the oldest subject was 33 years old. In the strenuous sports group, the mean values for cartilage volume were right talus (Ta_ri_) = 1.52 cm^3^ (range = 1.47–1.58 cm^3^); left talus (Ta_le_) = 1.49 cm^3^ (range = 1.41–1.57 cm^3^); right tibia (Ti_ri_) = 1.1 cm^3^ (range = 1.01–1.22 cm^3^); and left tibia (Ti_le_) = 1.06 cm^3^ (range = 0.96–1.22 cm^3^).

The cartilage volume average of the right talus (Ta_ri_) is 1.64 ± 0.280 cm^3^, the smallest volume (Ta_ri_) is 1.13 cm^3^, and the largest volume (Ta_ri_) is 2.14 cm^3^.

The cartilage volume average of the right tibia (Ti_ri_) is 1.25 ± 0.226 cm^3^, the smallest volume (Ti_ri_) is 0.86 cm^3^, and the largest volume (Ti_ri_) is 1.57 cm^3^.

The cartilage volume average of the left talus (Ta_le_) is 1.67 ± 0.310 cm^3^, the smallest volume (Ta_le_) is 1.16 cm^3^, and the largest volume (Ta_le_) is 2.10 cm^3^.

The cartilage volume average of the left tibia (Ti_le_) is 1.26 ± 0.215 cm^3^, the smallest volume (Ti_le_) is 0.96 cm^3^, and the largest volume (Ti_le_) is 1.53 cm^3^.

In the nonstrenuous sports group, the mean values for cartilage volume were Ta_ri_ = 1.84 cm^3^ (range = 1.48–2.12 cm^3^); Ta_le_ = 1.84 cm^3^ (range = 1.39–1.20 cm^3^); Ti_ri_ = 1.38 cm^3^ (range = 1.17–1.49 cm^3^); and Ti_le_ = 1.36 cm^3^ (range = 1.08–1.54 cm^3^).

In the nonathlete group, the mean values for cartilage volume were Ta_ri_ = 1.63 cm^3^ (range = 1.11–1.97 cm^3^); Ta_le_ = 1.65 cm^3^ (range = 1.11–2.06 cm^3^); Ti_ri_ = 1.29 cm^3^ (range = 0.85–1.58 cm^3^); and Ti_le_ = 1.30 cm^3^ (range = 0.90–1.55 cm^3^).

The observation and examination of the individual cartilage volumes by analysis of variance (ANOVA) yielded results of Ti_ri_ = 0.152, Ti_le_ = 0.117, Ta_ri_ = 0.183, and Ta_le_ = 0.287. Thus, there were no significant differences between the groups.

The effect sizes as measured by Cohen’s d were Ti_ri_ = 2.5, Ti_le_ = 2.2, Ta_ri_ = 1.9, and Ta_le_ = 1.6 in the strenuous sport versus nonstrenuous sport groups; Ti_ri_ = 0.8, Ti_le_ = 1.2, Ta_ri_ = 0.4, and Ta_le_ = 0.5 in the strenuous sport versus nonathlete groups; and Ti_ri_ = 0.3, Ti_le_ = 0.2, Ta_ri_ = 0.7, and Ta_le_ = 0.5 in the nonstrenuous sport versus nonathlete groups.

### 3.2. Device Comparison

A total of *n* = 14 subjects were examined on the 1.5 T device and *n* = 15 subjects on the 3.0 T device. The average age of all subjects (*n* = 15) was 26.07 ± 4.23 years. The youngest subject was 19 years old, and the oldest subject was 33 years old.

The assessment average of all subjects (*n* = 29) was 4.201 ± 0.815 (range = 2–5) for the bone–cartilage boundary (KKG), 4.043 ± 0.865 (range = 2–5) for the cancellous bone (Sp), 4.103 ± 0.900 (Range = 2–5), for the lateral ligamentous apparatus (lBa) at 3.655 ± 0.936 (Range = 2–5), and for the adipose tissue (Fg) at 3.379 ± 0.903 (Range = 2–5).

The mean score of the subject images on the 1.5 T device (*n* = 14) was 3.643 ± 0.745 (range = 2–5) for the cartilage–bone border (KKG), 3.429 ± 0.756 (range = 2–5) for the cancellous bone (Sp), 3.429 ± 0.756 (range = 2–4) for the medial ligamentous apparatus (mBa), 3.000 ± 0.784 (range = 2–4) for the lateral ligamentous apparatus (lBa), and 2.786 ± 0.802 (range = 2–4) for the adipose tissue (Fg).

The mean score of the subject images on the 3.0 T device (*n* = 15) was 4.733 ± 0.458 (range = 4–5) for KKG, 4.600 ± 0.507 (range = 4–5) for Sp, 4.733 ± 0.458 (range = 4–5) for mBa, 4.267 ± 0.594 (range = 3–5) for lBa, and 3.933 ± 0.594 (range = 3–5) for Fg.

The observation and examination of the individual assessment areas on the 1.5 T and 3.0 T MRI by the Wilcoxon matched-pair differences test yielded results of KKG = 0.002, Sp = 0.001, mBa = 0.001, lBa = 0.001, and Fg = 0.002. Thus, there were significant differences in the assessment of the 1.5 T MRI and 3.0 T MRI in all five evaluated areas.

## 4. Discussion

The largest average cartilage volume was found in the subjects of the nonstrenuous sport group, followed by the subjects of the nonathlete group. The smallest cartilage volume was found in the subjects of the strenuous sport group. However, the differences in cartilage volume were not significant. The corresponding observation of the effect intensity showed effects despite the small number of subjects.

Still, these results require careful interpretation as changes to lower limb cartilage following sports could be transient. There is evidence that cartilage could recover from sports stress [20].

In the comparison of the strenuous sport versus nonstrenuous sport groups, a large effect was found in all areas (Ti_ri_, Ti_le_, Ta_ri_, Ta_le_).

The comparison of the strenuous sport versus nonathlete groups showed a large effect in Ti_ri_ and Ti_le_, a small effect in Ta_ri_, and a medium effect in Ta_le_.

In the comparison of the nonstrenuous sport versus nonathlete groups, large effects were also found in all areas (Ti_ri_, Ti_le_, Ta_ri_, and Ta_le_). However, the small number of subjects should be noted; a tendency can be assumed, but these results are not generalizable. Lu L showed that strenuous sport could be a risk factor for the initiation of osteoarthritis [21]. There are clear influences of different sports on cartilage adaptations, such that running, swimming, ballet, and handball were not correlated with detrimental structural or molecular cartilage adaptation; instead, soccer, volleyball, basketball, weightlifting, climbing, and rowing showed signs of cartilage alteration that could be early predictive degeneration signs [22].

Furthermore, the inhomogeneity of the groups should be noted. The smallest cartilage volume was measured in the strenuous sports group, which included only women. In the nonstrenuous sport and nonathlete groups, both men and women were examined. Differences between male and female subjects were previously described by Faber et al. [3] in 2001. Spannow et al. [23] also showed a sex-specific difference in male and female adolescents by ultrasound examination in 2010. Regional and gender variations of ankle cartilage characteristics were found in dancers and non-dancers, with male dancers showing larger cartilage thickness than female dancers and non-dancers [24]. Furthermore, when examining the tibial articular cartilage of the knee joint, a 13% greater cartilage volume was measured in male subjects [25], which had a substantial influence on the study results.

The qualitative assessment by three radiologists with defined standards, who assessed the blinded images on a 5-point scale for sharpness or contrast of structures, showed an overall tendency in favor of the 3.0 T device. Naturally, within the framework of subjective perception of the quality of the images, a heterogeneous picture as well as a large interrater variance emerged. A clear over- or underassessment by the individual radiologists cannot be determined for the individual evaluation areas. For this reason, the significant differences in the assessment of the groups should not be overinterpreted. Our results of the superiority of the 3.0 Tesla investigations are in line with many other observations, resulting in the recommendation of the use of 3.0 Tesla scanners in the in vivo evaluation of distal joints [26].

Difficulties in the experimental arrangement occurred that partly influenced the results, since the coils could not be positioned outside the vacuum rail. The rail itself produced a reduced imaging quality, and the coil had to be built into the fixation apparatus. However, this reduced the stability of the fixation of the foot and resulted in significant tilting into a plantar-inflected position.

To optimize the foot position, buckles and reinforcement in the heel area should be considered, as this is where the greatest stress is exerted when the foot tips into a plantar-flexed position. In addition, it would be desirable to be able to position the coils outside the vacuum rail.

### Limitations of the Study

The overall limitation of the study results is the small number of included individuals. There was an unequal distribution between male and female subjects.

Furthermore, the quantitative assessment was performed using Volume Renderer from GE Healthcare. The hyaline articular cartilage was manually circumnavigated using a computer mouse. In this respect, there is a risk of inaccuracy. Furthermore, the separate measurement of tibial and talar cartilage also led to inaccuracies, as the joint space could not always be precisely delimited, resulting in double measurements in some areas. In repeated measurements, there was a measurement difference of 20.75% (M = 7.41%, SD = 4.04%).

## 5. Conclusions

The study showed no significant difference in the volume of hyaline articular cartilage in the upper ankle joint among the high-performance (strenuous) DFB athlete, high-performance DSV (non-strenuous) athlete, and nonathlete groups.

The 3.0 Tesla device offers significant advantages in image quality compared to the 1.5 Tesla device.

In further studies, female and male subjects should be considered in separate groups for each sport. Furthermore, a larger number of test subjects should be recruited.

To exclude a daytime difference in cartilage volume, further comparative studies at different times of the day would be useful.

## Figures and Tables

**Table 1 diagnostics-13-02750-t001:** Inclusion and exclusion criteria.

Inclusion Criteria	Exclusion Criteria
Healthy voluntees	Musculoskeletal disease
Informed consent	Osteochondral disease
Fitting into the groups	Previous fractures
	Current pain

## Data Availability

All data will be available from the authors upon request.

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
