# Peer review of "Ankle Joint MRI—Comparison of Image Quality and Effect of Sports-Related Stress"

_diagnostics, 2023, doi:10.3390/diagnostics13172750_

Round 1
Reviewer 1 Report
The objective of the paper is to evaluate the effects of stress with stimulating or reducing influences on cartilage growth in the ankle joint and to evaluate the image quality of a magnetic resonance imaging device by comparing two field strengths.
I suggest to indicate by * the corresponding author.
Abstract is well structured.
In the introduction, a good transposition is made in the research theme, and at the end, the objective of the study is formulated.
The study methodology is clearly described, as are the inclusion and exclusion criteria, but could be completed with a graphic diagram.
The results are presented synthetically and concisely indicating the measured values.
The discussion chapter interprets the results in relation to other achievements in the field, but I recommend a broader outline of the limitations of the study.
The conclusions are presented synthetically.
The references require some corrections for uniform editing.
Author Response
The objective of the paper is to evaluate the effects of stress with stimulating or reducing influences on cartilage growth in the ankle joint and to evaluate the image quality of a magnetic resonance imaging device by comparing two field strengths.
I suggest indicating by * the corresponding author.
Thank you for the suggestion. The corresponding author is indicated.
Abstract is well structured.
Thank you for this comment.
In the introduction, a good transposition is made in the research theme, and at the end, the objective of the study is formulated.
Thank you for this comment.
The study methodology is clearly described, as are the inclusion and exclusion criteria, but could be completed with a graphic diagram.
Thank you for this valuable proposal. A table has been added within the methods section (Table 1).
The results are presented synthetically and concisely indicating the measured values.
Thank you for this comment.
The discussion chapter interprets the results in relation to other achievements in the field, but I recommend a broader outline of the limitations of the study.
We have added some comments to the limitations of the study.
The conclusions are presented synthetically.
Thank you.
The references require some corrections for uniform editing.
The references have been corrected, thank you for the comment.
Please find attached two versions of the revised manuscript, one with mark-ups and one clean.
Sincerely
Professor Udo Rolle, MD, FEBPS, FRCS (Glasg)
Reviewer 2 Report
The conclusion - suggest mention the "groups"
Ethics? - no application? no ethics clearance? . Please add this. Declaration of helsinki?
Why was spss v18 used? We are presently on v28
Author Response
The conclusion - suggest mention the "groups"
Thank you for this comment, the groups have been included in the conclusions.
Ethics? - no application? no ethics clearance? . Please add this. Declaration of helsinki?
The ethical clearance reference has been added to the methods section.
Why was spss v18 used? We are presently on v28
We have used spss v18 as this was available at the time of the study.
Please find attached two versions of the revised manuscript, one with mark-ups and one clean.
Sincerely
Professor Udo Rolle, MD, FEBPS, FRCS (Glasg)
Reviewer 3 Report
Abstract
1. While the objectives of the study mentioned "evaluation of stress-related effects with stimulating or reducing influences on cartilage growth," the abstract does not explain the mechanism or hypothesis linking stress to cartilage growth.
2. The objectives talk about cartilage growth, but the results seem to focus on cartilage volume. Volume is not necessarily a direct indicator of growth - authors need to clarify if they are using volume as a proxy for growth and if so - they need to justify why and how this is a valid approach
3. The sample size is small (n=15) and it's not clear if it's sufficient to produce statistically significant results, especially for the ANOVA analysis across three groups. Additionally, it's unclear why the specific groups (DFB athletes, DSV athletes, non-athletes) were chosen. Are DFB athletes and DSV athletes representative of "strenuous" and "non-strenuous" sports respectively?
Introduction
1. The introduction discusses the changes in hyaline articular cartilage under stress but does not specify what kind of changes they are looking for - thickness, volume, morphology, or biochemical properties? It's necessary to define the changes being studied to set proper context.
2. The introduction cites several studies, but the discussion of the existing literature could be more comprehensive and organized. Instead of listing imaging techniques that can diagnose osteochondral lesions, the authors could compare and contrast these techniques to highlight the importance of their choice of MRI
3. It's important to clarify what is meant by "strenuous" and "non-strenuous" sports. This should ideally include specific criteria or examples, as well as a justification for these definitions.
Methods
1. The authors mention that this is a pilot study, but they don't explain why it's a pilot study (i.e., is it preliminary research for a larger, future study?).
2. While exclusion criteria are briefly mentioned, more details are needed. What specific musculoskeletal diseases or traumas were grounds for exclusion? What was the rationale behind these choices?
3. The authors should clearly explain how the 15 subjects were distributed across the three groups
4. The authors do not provide any detailed MRI parameters like field of view, slice thickness, repetition time, echo time, etc. These details are crucial for the reproducibility of the study.
5. More details are needed on how the blinded assessment was conducted. For instance, it's not clear how the images were randomized or how disagreements between the radiologists were resolved.
6. here is no mention of how the normality of data was checked before using the Wilcoxon and Kruskal-Wallis tests, which are non-parametric tests and assume non-normal data. Also, the authors should provide justification for the use of these specific tests and why they are appropriate for the data.
Results
Adequate
Discussions
1. The authors mention non-significant differences in cartilage volumes among groups. It might be more insightful to discuss why these results are not significant and how this non-significance could impact the interpretation of the study's results.
2. While the authors mention large, medium, and small effects, they do not clearly explain what this means in the context of their study. The implication of these effect sizes on the overall results and conclusions should be better explained.
3. While the authors discuss some limitations of their study, they could further discuss the implications of these limitations on the overall results. They could also highlight the strengths of their study and how it contributes to the existing literature.
4. The authors discuss some of the technical difficulties encountered during the study. It would be more informative if they provided more context on how these difficulties could have affected the results, and how they plan to overcome them in future studies.
Conclusions
Adequate
Author Response
- While the objectives of the study mentioned "evaluation of stress-related effects with stimulating or reducing influences on cartilage growth," the abstract does not explain the mechanism or hypothesis linking stress to cartilage growth.
Thank you for this very correct request. We looked at volume. The abstract has been corrected and changes of cartilage volume have been stated as main outcome measure.
- The objectives talk about cartilage growth, but the results seem to focus on cartilage volume. Volume is not necessarily a direct indicator of growth - authors need to clarify if they are using volume as a proxy for growth and if so - they need to justify why and how this is a valid approach.
The wording has been adapted; volume is the main interest.
- The sample size is small (n=15) and it's not clear if it's sufficient to produce statistically significant results, especially for the ANOVA analysis across three groups. Additionally, it's unclear why the specific groups (DFB athletes, DSV athletes, non-athletes) were chosen. Are DFB athletes and DSV athlete’s representative of "strenuous" and "non-strenuous" sports respectively?
The sample size has been checked before by biostatistical analysis and the group size has been proven to be appropriate.
Football is clearly a strenuous sport, swimming not. Thank you for that comment. The respective information has been added to the text.
Introduction
- The introduction discusses the changes in hyaline articular cartilage under stress but does not specify what kind of changes they are looking for - thickness, volume, morphology, or biochemical properties? It's necessary to define the changes being studied to set proper context.
The introduction has been modified; volume is the main interest.
- The introduction cites several studies, but the discussion of the existing literature could be more comprehensive and organized. Instead of listing imaging techniques that can diagnose osteochondral lesions, the authors could compare and contrast these techniques to highlight the importance of their choice of MRI.
Thank you for that comment. We have adapted the introduction section.
- It's important to clarify what is meant by "strenuous" and "non-strenuous" sports. This should ideally include specific criteria or examples, as well as a justification for these definitions.
The definition of strenuous vs. non-strenuous sports has been added to the text.
Methods
- The authors mention that this is a pilot study, but they don't explain why it's a pilot study (i.e., is it preliminary research for a larger, future study?).
This was a regular study to further clarify if these MRI techniques are suitable for cartilage volume investigations. We have removed “pilot”.
- While exclusion criteria are briefly mentioned, more details are needed. What specific musculoskeletal diseases or traumas were grounds for exclusion? What was the rationale behind these choices?
Specific exclusion criteria have been stated in the new table (requested by reviewer 1).
- The authors should clearly explain how the 15 subjects were distributed across the three groups.
Thank you for that comment. This required information has been added to the text.
- The authors do not provide any detailed MRI parameters like field of view, slice thickness, repetition time, echo time, etc. These details are crucial for the reproducibility of the study.
The required details on the MRI investigations have been added to the text.
- More details are needed on how the blinded assessment was conducted. For instance, it's not clear how the images were randomized or how disagreements between the radiologists were resolved.
The radiologists were blinded to the groups of included individuals.
- here is no mention of how the normality of data was checked before using the Wilcoxon and Kruskal-Wallis tests, which are non-parametric tests and assume non-normal data. Also, the authors should provide justification for the use of these specific tests and why they are appropriate for the data.
The respective explanation has been added to the methods section.
Results
Adequate
Discussions
- The authors mention non-significant differences in cartilage volumes among groups. It might be more insightful to discuss why these results are not significant and how this non-significance could impact the interpretation of the study's results.
Thank you for that comment. It is truly interesting, that we did not find significant changes in the cartilage volume between the groups. This is most probably due to the overall young age of the individuals.
- While the authors mention large, medium, and small effects, they do not clearly explain what this means in the context of their study. The implication of these effect sizes on the overall results and conclusions should be better explained.
Thank you for this comment. We have tried to explain further.
- While the authors discuss some limitations of their study, they could further discuss the implications of these limitations on the overall results. They could also highlight the strengths of their study and how it contributes to the existing literature.
Thank you for that comment. We have modified the discussion.
- The authors discuss some of the technical difficulties encountered during the study. It would be more informative if they provided more context on how these difficulties could have affected the results, and how they plan to overcome them in future studies.
We have added a respective paragraph to the limitation section.
Conclusions
Adequate
Please find attached two versions of the revised manuscript, one with mark-ups and one clean.
Sincerely
Professor Udo Rolle, MD, FEBPS, FRCS (Glasg)
Round 2
Reviewer 2 Report
Rephrase L98 - I don't think "probands" is the appropriate term in this context and "consent" must be spelt correctly
Author Response
Thank you for the comment.
We have rephrased the sentence and corrected the spelling.
